# Antiphospholipid Antibodies and Vascular Thrombosis in Patients with Severe Forms of COVID-19

**DOI:** 10.3390/biomedicines11123117

**Published:** 2023-11-22

**Authors:** Mirjana Zlatković-Švenda, Milica Ovuka, Manca Ogrič, Saša Čučnik, Polona Žigon, Aleksandar Radivčev, Marija Zdravković, Goran Radunović

**Affiliations:** 1Institute of Rheumatology Belgrade, 11000 Belgrade, Serbia; g.radunovic47@gmail.com (G.R.); aleksandar.radivcev@gmail.com (A.R.); 2Faculty of Medicine, University of Belgrade, 11000 Belgrade, Serbia; sekcija.kardioloska@gmail.com; 3Faculty of Medicine Foča, University of East Sarajevo, 73300 Foča, Bosnia and Herzegovina; 4Clinical Hospital Center Pančevo, 26101 Pancevo, Serbia; movuka98@gmail.com; 5Institute for Cardiovascular Diseases Dedinje, 11000 Belgrade, Serbia; 6Department of Rheumatology, University Medical Centre Ljubljana, 1000 Ljubljana, Slovenia; manca.ogric@kclj.si (M.O.); sasa.cucnik@kclj.si (S.Č.); polona.zigon@guest.arnes.si (P.Ž.); 7Faculty of Pharmacy, University of Ljubljana, 1000 Ljubljana, Slovenia; 8Faculty of Mathematics, Natural Sciences and Information Technologies, University of Primorska, 6000 Koper, Slovenia; 9Clinical Hospital Bežanijska Kosa, 11000 Belgrade, Serbia

**Keywords:** COVID-19, antiphospholipid syndrome, vascular thrombosis, antiphospholipid antibodies, anticardiolipin antibodies (aCL), anti-β2-glycoprotein I antibodies (anti-β2GPI), anti-phosphatidylserine-prothrombin (aPS/PT) antibodies

## Abstract

Antiphospholipid antibodies (aPLA) are a laboratory criterion for the classification of antiphospholipid syndrome (APS) and are known to cause clinical symptoms such as vascular thrombosis or obstetric complications. It is suggested that aPLA may be associated with thromboembolism in severe COVID-19 cases. Therefore, we aimed to combine clinical data with laboratory findings of aPLA at four time points (admission, worsening, discharge, and 3-month follow-up) in patients hospitalized with COVID-19 pneumonia. In 111 patients with COVID-19 pneumonia, current and past history of thrombosis and pregnancy complications were recorded. Nine types of aPLA were determined at four time points: anticardiolipin (aCL), anti-β2-glycoprotein I (anti- β2GPI), and antiphosphatidylserine/prothrombin (aPS/PT) of the IgM, IgG, or IgA isotypes. During hospitalization, seven patients died, three of them due to pulmonary artery thromboembolism (none were aPLA positive). Only one of the five who developed pulmonary artery thrombosis was aPLA positive. Out of 9/101 patients with a history of thrombosis, five had arterial thrombosis and none were aPLA positive at admission and follow-up; four had venous thrombosis, and one was aPLA positive at all time points (newly diagnosed APS). Of these 9/101 patients, 55.6% were transiently aPLA positive at discharge only, compared to 26.1% without a history of thrombosis (*p* = 0.041). Patients with severe forms of COVID-19 and positive aPLA should receive the same dose and anticoagulant medication regimen as those with negative aPLA because those antibodies are mostly transiently positive and not linked to thrombosis and fatal outcomes.

## 1. Introduction

Antiphospholipid syndrome (APS) is the most prevalent type of acquired thrombophilia [1], associated with thromboses of venous and arterial blood vessels and therefore linked to cardiovascular events, stroke, and fatal outcomes. Classification of APS goes through the revised Sapporo criteria that require at least one clinical criterion, vascular thrombosis or pregnancy morbidity, one laboratory criterion, and the presence of any of the antiphospholipid antibodies (aPLA) confirmed at least 12 weeks apart [2]. aPLA comprise IgG and IgM isotypes of lupus anticoagulant (LA), anticardiolipin (aCL), and/or anti-β2-glycoprotein I (anti-β2GPI) antibodies [2]. In patients where the LA test shows methodological flaws, such as those on anticoagulation therapy, antiprothrombin antibodies (aPS/PT) can be performed in addition to aCL and anti-β2GPI, as together they have shown to cover the majority of LA activity [3,4]. Some studies have reported a strong correlation between aPS/PT positivity and LAC, showing positive aPS/PT in 77–100% of single LAC-positive cases [5]. Furthermore, aPS/PT have shown an important correlation with APS clinical manifestations [6,7,8]. However, not all single LA positive patients are aPS/PT positive [9].

The incidence rate of APS in women and men is reported differently. According to one study, it is 5:1 [10], while another study has found a similar ratio between men and women [11]. Women are vulnerable to APS because aPLA are found to be one of the main reasons for recurrent pregnancy loss (spontaneous abortion), besides uterine factors and congenital abnormalities [12], and also for thrombosis (arterial, venous, or microthrombosis) and pregnancy complications [13]. In addition, aCL are one of the main reasons for blood vessel thrombosis (arterial, venous, or microthrombosis). That is the reason why treatment takes the lifelong oral anticoagulation or antiplatelet therapy, which in most cases prevents recurrence of thrombosis [14]. 

It was discovered recently that the thrombogenic potential of aPLA could be enhanced with the presence of infection or severe inflammation, which is the so-called “two-trigger hypothesis” [15]. According to this theory, it was hypothesized that COVID-19 could play the role of the “second trigger”.

A worldwide pandemic (COVID-19) has a highly heterogeneous clinical course, including pulmonary embolism, deep vein thrombosis, and stroke in severe forms of the disease [16,17,18]. Since some patients with COVID-19 were shown to have positive aPLA [19,20], it was hypothesized that these antibodies could add to the vascular complications.

Moreover, the often fatal catastrophic variant of APS shares some similarities with diffuse coagulopathy that could be seen in patients with COVID-19 [13]. Numerous studies have found high rates of aPLA in patients hospitalized with COVID-19; either they had thrombosis or not [17,21,22]. The importance of aPLA in promoting thrombosis during COVID-19 is still ambiguous.

The aim of this study was to evaluate the laboratory results of the erythrocyte sedimentation rate (ESR), C-reactive protein (CRP), complete blood cell count (CBC) tests, D-dimer, ferritin, troponin, procalcitonin, lactate dehydrogenase (LDH), and antiphospholipid antibodies (aPLA): aCL, anti-β2GPI, and aPS/PT of the IgG, IgM, and IgA isotypes) obtained at four time points (hospital admission, worsening of COVID-19 disease, discharge from the hospital, and at 3-month follow-up) and compare the laboratory results of aPLA with clinical data on arterial/venous thrombosis and pregnancy complications before (according to personal anamnesis) and during hospitalization in patients with severe forms of COVID-19 hospitalized due to pneumonia.

## 2. Materials and Methods

### 2.1. Patients and Data Extraction

The study had randomly enrolled patients over 18 years of age with COVID-19 pneumonia who were admitted to the Intensive Care Unit (ICU) of the General Hospital Pančevo, Serbia, from March 2021 to May 2021. Involved patients had either antigens or positive PCR tests at admission and were all confirmed with positive PCR testing during hospitalization.

Exclusion criteria were history of autoimmune diseases, APS, connective tissue diseases (systemic lupus erythematosus, systemic sclerosis, Sjögren’s disease, vasculitis, dermatomyositis) or inflammatory arthritis (rheumatoid arthritis and spondyloarthritis). In the ICU, continuous oxygenation was performed, and continuous noninvasive monitoring with pulse oximetry, electrocardiography, and monitoring of blood pressure, pulse, and temperature were performed every 4 h. 

The history of clinical manifestations attributable to APS was obtained from medical records and also monitored during the period of observation. The study was conducted within the framework of the Slovenian national project P3-0314 according to the guidelines of the Declaration of Helsinki and was approved by the Ethics Committee of the Republic of Slovenia (#0120-7/2019/5, #0120-422/2020/6, and #0120-113/2021/4), the Ethics Committee of the Republic of Serbia (132/2, 14 January 2021), and the Ethics Committee of the General Hospital of Pančevo (#01-1492/21). Informed consent was obtained from all subjects participating in the study.

### 2.2. Study Endpoints and Follow-Up 

Patients were monitored for clinical and laboratory features during hospitalization, i.e., at admission, at worsening of the COVID-19 disease (defined as cytokine storm, connection of the patient to the respirator, and utilization of the anti-interleukin (IL)-6 drug Tocilizumab), at hospital discharge, and 3 months after discharge. 

### 2.3. Blood Sampling and Antiphospholipid Antibody Measurement

At each time point, blood samples were routinely collected for determination of ESR, CRP, CBC, D-dimer, ferritin, troponin, procalcitonin, LDH; they were determined at the Pancevo hospital.

Also, at each time point, serum samples for aPLA testing (aCL, anti-β2GPI, and aPS/PT of IgG, IgM, and IgA isotypes) were collected following the previously established protocol. Samples were allowed to clot for 30 to 60 min and then centrifuged at 1500× *g* for 15 min to separate clot and serum. The serum was transferred to 2 Protein LoBind Tubes (Eppendorf, Hamburg, Germany) and stored at −80 °C until shipment on dry ice to prevent thawing. aPLA were determined at the Department of Rheumatology, University Medical Centre Ljubljana (UMCL), Slovenia, using in-house enzyme-linked immunosorbent assays (ELISA) according to previously established protocols [23,24] in compliance with the international guidelines [25] protocol. Briefly, aCL were measured according to the protocol first described in 1997 [23] and later repeatedly evaluated [26,27]. anti-β2GPI were measured by in-house ELISA [28] and evaluated by the European Forum for aPL [29]. aPS/PT were measured with in-house ELISA first described in 2011 [24] and later repeatedly evaluated [6,30,31]. Values above the 99th percentile of the healthy control population were regarded as positive, i.e., for aCL ≥ 11AU, for anti-β2GPI ≥ 2 AU, and for aPS/PT ≥ 5 AU.

### 2.4. Statistical Analyses

Frequencies (mean, standard deviation [SD], interquartile range [IQR], median) were used for data description. A one-sample runs test was used for ascertaining if the sequence of values is random. The one-sample Kolmogorov–Smirnov test was used to test normality of distribution. For admission/discharge/follow-up comparisons, for normally distributed continuous variable paired samples, a t test was used, and for non-normally distributed variables, the Wilcoxon signed-ranks test was used. For aPLA positivity testing on a dichotomous scale, the McNemar test was used. Pearson’s chi-square test was used for comparisons between different groups. All tests were performed by SPSS (version 21; IBM, Armonk, NY, USA). A p-value of 0.05 or less was considered statistically significant.

## 3. Results

### 3.1. Patient Baseline Characteristics

At baseline, 111 SARS-CoV-2-positive patients admitted to General Hospital Pančevo for COVID-19 pneumonia from March 2021 to May 2021 were included. Later, three patients declined to provide their consent to take part in the study (Figure 1). The remaining 108 patients (67% men, mean ± SD age 59.4 ± 12.4 years (IQR 16 years)) were monitored in the intensive care unit (ICU) and received treatment for pneumonia according to the established COVID-19 protocol, which includes administration of antibiotics, corticosteroids, anticoagulants, and certain medications for comorbidities; patients who were hypoxic received oxygen support. Patients had hypertension (47.2%), hyperlipidemia (41.6%), diabetes mellitus (23.9%), and angina pectoris (10.6%) as comorbidities. During hospitalization, the condition of 13 patients worsened. Three patients died before laboratory test results were available. Of the remaining ten patients, six were treated with Tocilizumab and four with high oxygen flow. Despite therapy, four additional patients died. Follow-up, performed 3 months after discharge, was achieved in 91 patients. 

### 3.2. Laboratory Values at Four Time Points

Table 1 presents the laboratory parameters measured at four different time points. ESR and ferritin levels were much lower at hospital release than at admission, but this change was clinically insignificant because they remained above the reference range; the results did not normalize to the end of the follow-up. In contrast, CRP and LDH levels decreased rapidly and returned to normal at discharge. The small improvements in hemoglobin, platelets, hematocrit, and erythrocyte number at the 3-month follow-up were clinically insignificant. During hospitalization and at follow-up, procalcitonin levels remained below the reference range.

The aPLA levels were significantly higher at hospital discharge than at admission and follow-up, particularly aCL IgG, aCL IgM, and anti-β2GPI IgG. (Table 1). 

Positivity of aPLA (values above the reference range) in COVID-19 patients hospitalized in the ICU due to pneumonia at four time points is given in Table 2.

### 3.3. Specific Subgroups of Patients

#### 3.3.1. Patients Who Died during Hospitalization

Seven patients (mean age 65.8 years, range 52 to 77 years) passed away during hospitalization, three of them from pulmonary artery thromboembolism, two from cytokine storm, and two on the respirator, although the exact cause of death was unknown in their case. Of those who died, two were treated with tocilizumab due to cytokine storm, two required respirators, and four received high oxygen flow. None of these patients tested positive for aPLA on admission (Figure 2).

#### 3.3.2. Patients Who Experienced Thrombosis during Hospitalization

Five patients experienced pulmonary artery thrombosis during hospitalization, one had positive aPLA at all time check-out points and was therefore diagnosed with APS, while the others tested negative (Figure 3).

#### 3.3.3. Patients with a History of Thrombosis

A history of thrombosis was found in 9/101 subjects, of whom five had arterial thrombosis (coronary and cerebral arteries) and four had venous thrombosis. Of the patients with arterial thrombosis, none had a positive aPLA either at admission or at follow-up. On the other hand, among patients who had a history of venous thrombosis, one tested positive for aPLA at all time points and was diagnosed with APS. The other three patients with VT had transiently positive aPLA values (two at hospital discharge and one at admission).

Of nine individuals who had a history of arterial or venous thrombosis, 5/9 (55.6%) had transiently positive aPLA results only at hospital discharge, whereas a much smaller proportion of patients without a history of thrombosis tested transient aPLA positive only at hospital discharge: 22/92 (23.9%) (*p* = 0.041) (Figure 4).

#### 3.3.4. Patients with a History of Pregnancy Morbidity

Regarding history of pregnancy morbidity, two patients had an abortion after the 10th week of gestation, but none of them tested positive for aPLA at any time point. In our cohort, there were no patients who had delivered before the 34th week of gestation or who had 3 or more abortions before the 10th week of gestation.

## 4. Discussion

For patients hospitalized with COVID-19 pneumonia, we have combined laboratory results for nine types of aPLA at four time points (admission, worsening, discharge, and 3-month follow-up) with clinical information on vascular thrombosis and pregnancy complications. 

The current longitudinal investigation on the relationship between thrombosis and antiphospholipid antibodies (aPLA) in COVID-19 patients with severe manifestations has several obvious long-term applicable implications.

First, with regard to COVID-19 severity, thrombosis of the pulmonary artery and aPLA, none of the patients hospitalized for COVID-19 pneumonia in our study who passed away had positive aPLA at the time of admission, albeit three of them had thrombosis of the pulmonary artery. In addition, only one of the five patients who experienced pulmonary artery thromboembolism while being treated in the hospital had aPLA positivity at all time checkpoints and was therefore given a diagnosis of APS; the other four patients tested aPLA negative. Second, with regard to thrombosis of other organs, COVID-19 severity and aPLA, two patients with COVID-19 pneumonia developed arterial thrombosis during hospitalization—one developed cerebral artery thrombosis and was positive for aCL IgG, anti-β2GPI IgM, and aPS/PT IgM on admission. The other patient developed coronary artery thrombosis with myocardial infarction and was aPLA negative on admission. Additionally, five patients experienced microthrombosis while they were hospitalized; one had pulmonary microthrombosis and was aPLA positive (aCL IgG and anti-β2GPI IgM) at admission, while the other four were aPLA negative (they experienced pulmonary microthrombosis and coronary artery microthrombosis). None of the hospitalized patients who experienced arterial thrombosis or microthrombosis had a transiently positive aPLA at discharge compared to admission and follow-up.

Third, with regard to thrombosis in a patient‘s personal history, COVID-19 severity and aPLA when discharged from the hospital was compared to admission and 3-month follow-up; patients who had prior thrombosis had significantly higher transient increases in aPLA (aCL IgG, aCL IgM, and anti-2GPI IgG) than patients who did not have thrombosis. 

The significance of aPLA in initiation of thrombotic events in COVID-19 patients remains unclear. Elevated levels of aPLA are found in many COVID-19 patients, either with or without thrombosis [8,32,33]. Although it has been hypothesized that aPLA may help cause the hypercoagulability condition, it is also possible that it is an epiphenomenon linked to the COVID-19 infection. Hypercoagulability states may be a result of aPLA, but it is also possible that it could be caused by the COVID-19 infection itself. In support of this, apart from a higher prevalence of aPLA, a recent study has confirmed prothrombotic potential of peripheral blood in patients hospitalized with COVID-19, showing decreased levels of natural plasma anticoagulants (protein C 17% lower, free protein S 22% lower) and unchanged thrombin generation capacity when compared to controls. The prothrombotic shift in COVID-19 patients’ blood may be a factor in their higher risk of thromboembolism [34]. 

On the other hand, whether aPLA are permanently or only temporarily elevated during COVID-19 infection and which types of them have not been fully investigated yet. Given that the most difficult forms of COVID-19 infection are linked to cardiovascular and neurological consequences, the dosage and duration of anticoagulant therapy could be determined by the extent of the aPLA present in these patients.

Patients with and without hypertension, hyperlipidemia, diabetes mellitus, angina pectoris, and myocardial infarction were equally found to have transiently positive aPLA in our sample. In addition, patients with and without hypertension, hyperlipidemia, diabetes mellitus, angina pectoris, and myocardial infarction did not significantly differ in terms of the frequency of arterial, vein, or microthrombosis during hospitalization.

According to reports, SARS-CoV-2 infection has a disproportionately higher promoting effect on the patient’s coagulation system. The endothelial dysfunction caused by SARS-CoV-2 and the generation of cytokines and growth factors are anticipated to have a major impact on platelet activation, coagulopathy, and venous thromboembolism [35].

In order to follow the COVID-19 disease’s various consequences, numerous studies have investigated the induction of aPLA in this disease. It has been documented that COVID-19 infection may be the only trigger for thrombosis in multiple organs [21]. Furthermore, the infusion of IgG isolated from COVID-19 patient serum accelerated venous thrombosis in two animal models [8]. 

In the present study, changes observed in aPLA positivity were mostly due to aCL, which were considerably lower at hospital admission compared to discharge, whereas anti-β2GPI and aPS/PT did not show that difference. Significantly higher transient elevations of aCL IgG, aCL IgM, and anti-β2GPI IgG were found at hospital discharge (compared with admission and follow-up) for patients who had COVID-19 pneumonia and thrombosis in their personal history when compared to those without previous thrombosis. This refers to an increase in the numbers of positive assays, e.g., for patients with a history of thrombosis, levels of different aPL increased above the cut-off. However, none of these patients experienced thrombosis while hospitalized. 

According to published data, 50% of patients hospitalized with COVID-19 have at least transitory positivity of aPLA antibodies, and some researchers hypothesized that these autoantibodies may be harmful [8,19,20]. It is challenging to draw conclusions about the link and causal dependency between aPLA positivity and COVID-19 disease outcomes, such as thrombosis or mortality, because different research have produced conflicting results [22,36]. 

It was reported that thrombotic events occurred in 116 of the 163 individuals with virus-associated aPLA, although the clinical implications of transient virus-associated aPLA antibodies have not yet been fully clarified [37]. 

It has been discovered that COVID-19’s difficult forms are highly correlated with total IgA and specific subtypes of IgA that belong to aPLA [20]. It is also believed that a rise in IgA subtypes of aPLA causes thrombosis. However, none of the patients in our study who experienced arterial, venous, or microthrombosis or pulmonary artery thrombosis exhibited an elevated IgA subtype of any aPLA type (aCL, anti-2GPI, or aPS/PT).

There were no pregnant women in our cohort. Two females had an abortion after the 10th week of gestation in their personal anamnesis, and none of them had positive aPLA at any time point. In the up-to-date literature, only one study reported a stillborn in a pregnant woman with positive aCL during the COVID-19 pandemic [38], but we have found no data about pregnancy complications in personal anamnesis and aPLA positivity in patients with SARS-CoV-2 infection. 

Positive aPLA were previously found in severely ill individuals, both those with and without the COVID-19 infection, and there was an independent association between presence of aCL IgG and disease severity, regardless of the COVID-19 status. Furthermore, there were no discernible differences in platelet counts, platelet-to-neutrophil ratios, or the need for therapeutic anticoagulation in those with aCL IgG positivity [19].

In comparison to admission and follow-up, our study has demonstrated a temporary rise in aPLA in COVID-19 patients with pneumonia at the time of hospital discharge. Moreover, transiently positive aPLA were significantly more prevalent in patients who had previous thrombosis (55.6%) than in patients without thrombosis (26.1%).

While 50–70% of COVID-19 patients tested positive for at least one aPL, according to certain investigations, they did not experience any thrombotic events [39]. Despite the prior descriptions of connections between elevated aPL and severe COVID-19 [40], these observations are only correlations and do not prove causality [40]. Since the majority of patients’ aPL are transient, the elevated levels in severe COVID-19 are more likely due to the illness condition worsening, and the concomitant further elevation of epiphenomenal antibodies is not necessarily remarkable. 

The main limitation of our study is that the LA measurement could not be performed for an important reason. This is because all of our patients were already receiving anticoagulation therapy for COVID-19 pneumonia (dosing of low molecular weight heparin Fraxiparine ranged from 2 × 0.6 mL to 2 × 0.9 mL daily, depending on the level of D-dimer), and anticoagulants may interfere with the tests used to detect LA, occasionally leading to false-positive or false-negative LA. Although it is believed that when given with neutralizers, LMWH would not alter the LA testing, we were unable to employ neutralizers due to their effectiveness being restricted to specific LMWH doses (0.8–1.0 U/mL), which typically cover only preventive dosing [41]. Utilizing LMWH can prolong significant clotting tests and APTT length depending on their anti-Fxa/FIIa ratio, which may interfere with the LA detection [42,43]. The LA Scientific and Standardization Committee guidelines published by the International Society on Thrombosis and Haemostasis conclude that the detection of LA during anticoagulation remains a challenge [43]. Even the most advanced centers, let alone general hospitals, face these problems. To overcome problems with the LA testing in patients taking anticoagulants, some authors suggest that measurement of aPS/PT may be the closest substitute [44,45,46]. Additionally, aPS/PT may be helpful in patients with incomplete antibody profiles due to the methodological limitations of immunological and clotting assays [45]. The main drawback of this study is that some writers continue to question whether aPS/PT has additional value in APS diagnosis or assessment of risks when compared to current criteria [47].

The major strength of this study is the longitudinal comparison of clinical data regarding thromboses (both actual and in personal history) with aPLA findings during hospitalization and 3 months later at follow-up.

## 5. Conclusions

Despite the fact that vascular thrombosis in severe forms of COVID-19 infection were predicted to be linked with aPLA, all individuals with pneumonia in our cohort who passed away were aPLA negative. 

The novelty of this study is that among patients hospitalized for COVID-19 pneumonia, those with a history of arterial or venous thrombosis in more than 50% had positive aPLA at hospital discharge, which was later found to be negative at the 3-month follow-up. Also, of the patients who developed arterial thrombosis or microthrombosis during hospitalization, none had a transiently positive aPLA at hospital discharge, compared with admission and follow-up. This finding may help us in recommending anticoagulation therapy after hospitalization. Patients with severe COVID-19 who have positive aPLA should receive the same anticoagulation therapy as those without aPLA, since those antibodies are typically transiently positive (they mostly become negative within 3 months) and not connected with the new occurrence of thromboses.

## Figures and Tables

**Figure 1 biomedicines-11-03117-f001:**
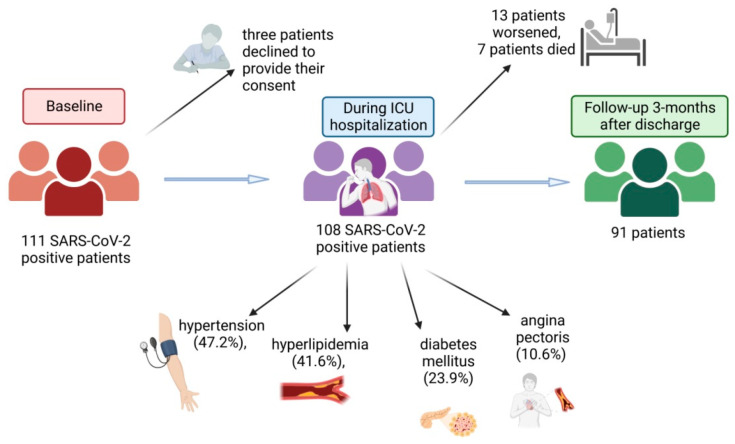
Study population and patient characteristics. (Created with BioRender.com).

**Figure 2 biomedicines-11-03117-f002:**
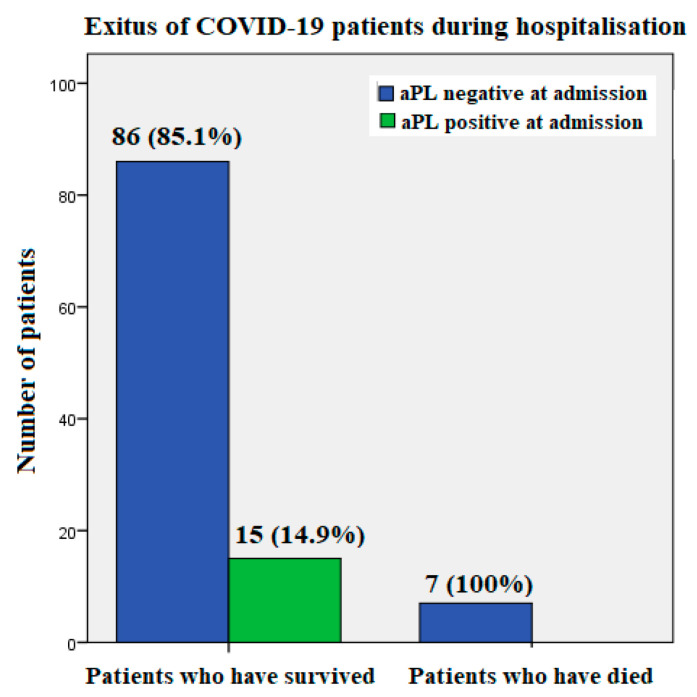
Exitus and the presence of antiphospholipid antibodies (aPL) at admission in SARS-CoV-2-positive patients hospitalized with COVID-19 pneumonia.

**Figure 3 biomedicines-11-03117-f003:**
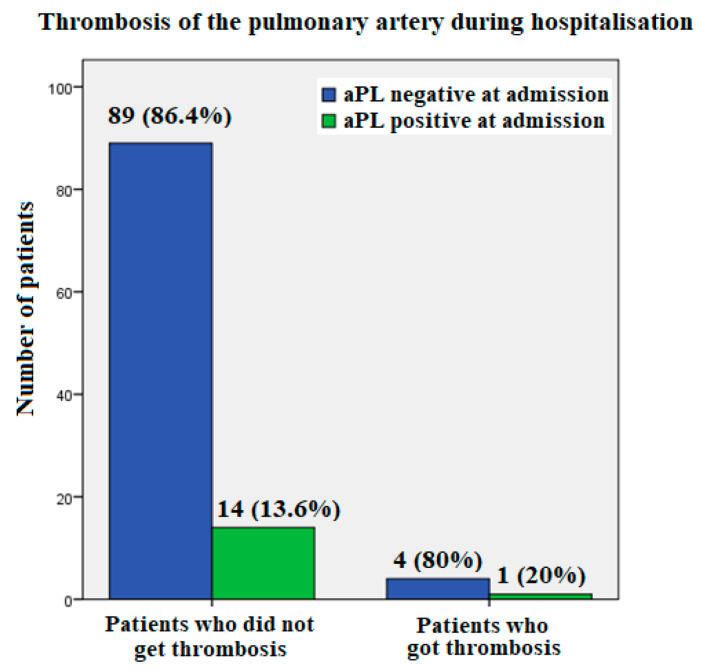
Pulmonary artery thrombosis and the presence of antiphospholipid antibodies (aPLA) in SARS-CoV-2-positive patients hospitalized with COVID-19 pneumonia.

**Figure 4 biomedicines-11-03117-f004:**
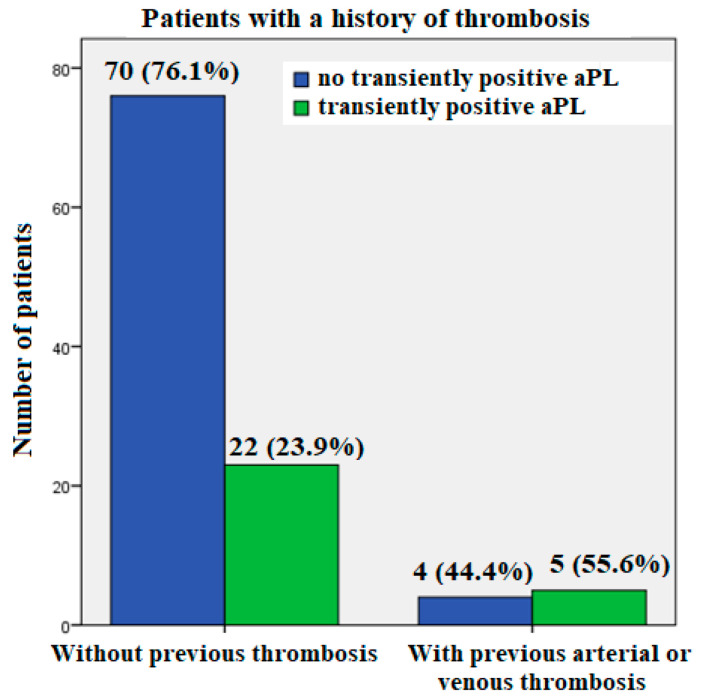
SARS-CoV-2 positive patients with COVID-19 pneumonia: transiently positive antiphospholipid antibodies (aPLA) at hospital discharge (negative upon hospital admission and at 3-month follow-up) in patients with and without arterial/ venous thrombosis in their medical history.

**Table 1 biomedicines-11-03117-t001:** Laboratory parameters of COVID-19 patients hospitalized in the ICU due to pneumonia at the time of admission, worsening, discharge from the hospital, and at 3-month follow-up.

		Normal Range	Admission (*n* = 107)	Worsening (*n* = 10)	Discharge (*n* = 101)	Follow-Up (*n* = 91)	Admission/ Discharge	Admission/Follow-Up
		Mean ± SD /Median (IQR) ^†^					*p* - Value	
HAEMATOLOGY	Erythrocyte sedimentation rate (mm/h)	0–15	50.2 (22.4)	61.5 (8.5)	28 (11.2)	12.2 (11.9)	<0.001 *	<0.001 *
	C- reactive protein (mg/L) **^†^**	<5.0	91.8 (93.7)**^†^**	123.4 (194.5) **^†^**	3.6 (5.9) **^†^**	2.8 (4.7) **^†^**	<0.001 *	<0.001 *
	Erytrocytes (×10^12^/L)	4.50–6.30	4.51 (0.5)	4.3 (0.4)	4.6 (1.3)	4.8 (0.5)	0.24	<0.001 *
	Haemoglobin (g/L)	140–175	132.3 (15.7)	129.1 (15.6)	132.2 (13.9)	139 (14.6)	0.98	<0.001 *
	Haematocrit (%)	0.400–0.520	0.4 (0.04)	0.4 (0.04)	0.4 (0.04)	0.4 (0.05)	0.35	<0.001 *
	Leucocytes (×10^9^/L)	4.40–11.50	6 (2.5)	8.4 (5)	10.8 (4.2)	7.1 (2.2)	<0.001 *	<0.001 *
	Neutrophils (×10^9^/L)	2.20–8.05	3.12 (0.78)	5.92 (3.48)	7.66 (2.18)	3.65 (0.77)	<0.001 *	0.02 *
	Lymphocytes (×10^9^/L)	1.10–4.60	0.75 (0.44)	0.55 (0.26)	0.87(0.05)	2.4 (0.69)	<0.001 *	<0.001 *
	Platelets (×10^9^/L)	150–400	214.9 (78)	224.3 (87.1)	315.6 (105.7)	246 (55.4)	<0.001 *	0.01 *
BIOCHEMISTRY	D-dimer (mg/L) **^†^**	<0.5	0.71 (0.81) **^†^**	1.18 (3.15) **^†^**	0.50 (0.55) **^†^**	0.39 (0.34) **^†^**	0.04 *	<0.001 *
	Ferritin (µg/L)	4.63–204.00	997 (664.8)	1523.3 (420.2)	692 (533.2)	195.3 (185)	<0.001 *	<0.001 *
	Troponin (ng/L) **^†^**	0–11.6	5.91 (7.44) **^†^**	25.93 (23.47) **^†^**	3.19 (5.47) **^†^**	2.82 (2.36) **^†^**	<0.001 *	<0.001 *
	Procalcitonin (ng/mL) **^†^**	<0.08	0.07 (0.09) **^†^**	0.12 (0.22) **^†^**	0.02 (0.01) **^†^**	0.03 (0.02) **^†^**	<0.001 *	<0.001 *
	LDH (U/L)	230–480	666.8 (282)	1107.5 (487.8)	394 (111.2)	370.3(65.9)	<0.001 *	<0.001 *
ANTIPHOSPHOLIPID ANTIBODIES	aCL IgG (AUG)	≤10	4.49 (6.22)	7.00 (7.90)	9.13 (9.54)	5.99 (5.19)	<0.001 *	<0.001 *
	aCL IgM (AUM)	≤10	3.34 (3.77)	4.0 (1.79)	9.92 (12.10)	3.21 (6.53)	<0.001 *	<0.001 *
	aCL IgA (AUA)	≤10	1.71 (2.66)	1.71 (2.66)	1.65 (1.79)	0.45 (0.87)	0.73	<0.001 *
	anti-β2GPI IgG (AUG)	≤1	0.10 (0.31)	0.14 (0.36)	0.44 (1.42)	0.40 (0.99)	0.002 *	<0.001 *
	anti-β2GPI IgM (AUM)	≤1	0.04 (0.24)	0.00 (0.00)	0.10 (0.41)	0.20 (0.88)	0.32	0.07
	anti-β2GPI IgA (AUA)	≤1	0.23 (0.75)	0.21 (0.43)	0.25 (0.83)	0.20 (0.62)	0.81	0.37
	aPS/PT IgG (AUG)	≤4	1.03 (0.42)	1.21 (0.69)	0.93 (0.59)	1.22 (0.51)	0.07	<0.001 *
	aPS/PT IgM (AUM)	≤4	0.78 (3.60)	3.50 (12.26)	0.99 (4.52)	0.98 (4.71)	0.66	0.97
	aPS/PT IgA (AUA)	≤4	1.54 (1.19)	2.14 (1.23)	1.52 (1.22)	0.82 (1.24)	0.88	<0.001 *

* *p* < 0.05; *n*, number of patients; LDH, *lactate dehydrogenase*; AU, arbitrary units; aCL, anticardiolipin antibodies; anti-β2GPI, anti-β2-glycoprotein I antibodies; aPS/PT, anti-phosphatidylserine-prothrombin antibodies **^†^** For data sets with outliers, results are given as median (IQR); IQR, interquantile range.

**Table 2 biomedicines-11-03117-t002:** Antiphospholipid antibodies positivity in COVID-19 patients hospitalized in the ICU due to pneumonia at the time of admission, worsening, discharge from the hospital, and at 3-month follow-up.

	Admission (*n* = 107)	Worsening (*n* = 10 ) **	Discharge (*n* = 101)	Follow-Up (*n* = 91)	Admission/ Discharge	Admission/Follow-Up
	Number (%)				*p* -Value	
aCL IgG	9 (8.4)	3 (21.4)	28 (27.7)	13 (14.3)	<0.001 *	0.28
aCL IgM	4 (3.7)	0 (0.0)	23 (22.8)	6 (6.6)	<0.001 *	0.71
aCL IgA	3 (2.8)	0 (0.0)	1 (1)	0 (0.0)	0.32	0.08
anti-β2GPI IgG	0 (0.0)	0 (0.0)	7 (6.9)	5 (5.5)	0.01 *	0.02 *
anti-β2GPI IgM	1 (0.9)	0 (0.0)	2 (2)	4 (4.4)	1.00	0.16
anti-β2GPI IgA	4 (3.7)	0 (0.0)	4 (4)	4 (4.4)	1.00	1.00
aPS/PT IgG	0 (0.0)	0 (0.0)	0 (0.0)	0 (0.0)	1.00	1.00
aPS/PT IgM	5 (4.7)	1 (7.1)	5 (5)	4 (4.4)	1.00	0.32
aPS/PT IgA	1 (0.9)	0 (0.0)	2 (2)	1 (1.1)	0.32	1.00
aPLA positivity (9) ^1^	16 (14.8)	2 (20.0)	48 (47.5)	24 (26.4)	<0.001 *	<0.001 *
aCL positivity (3) ^2^	12 (10.8)	2 (20.0)	42 (41.6)	18 (19.8)	<0.001 *	0.33
anti- β2GPI positivity (3) ^3^	5 (4.5)	0 (0.0)	12 (11.9)	11 (12.1)	0.11	0.18
aPS/PT positivity (3) ^4^	4 (3.7)	1 (10.0)	6 (5.9)	4 (4.4)	0.50	1.00
Double positivity ^5^	1 (0.9)	1 (0.9)	9 (8.9)	5 (5.5)	0.02 *	0.10
Triple positivity ^6^	2 (1.9)	0 (0.0)	1 (1.0)	2 (2.2)	0.32	0.32

* *p* < 0.05; *n*, number of patients; ** 10 patients worsened, but some of them more than once, so there were 14 occasions of worsening, and the percentages are given accordingly; aPLA, antiphospholipid antibodies; aCL, anticardiolipin antibodies; anti-β2GPI, anti-β2-glycoprotein I antibodies; aPS/PT, anti-phosphatidylserine-prothrombin antibodies; ^1^ aPLA positivity, at least 1 of 9 evaluated antiphospholipid antibodies above the reference range; ^2^ aCL positivity, at least 1 out of 3 aCL above the reference range (IgG, IgM, IgA); ^3^ anti-β2GPI positivity, at least 1 out of 3 anti-β2GPI above the reference range (IgG, IgM, IgA); ^4^ aPS/PT positivity, at least 1 out of 3 aPS/PT above the reference range (IgG, IgM, IgA); ^5^ at least 1 in 2 out of 3 classes of aPLA positive: aCL, anti-β2GPI, aPS/PT; ^6^ at least 1 in 3 out of 3 classes of aPLA positive: aCL, anti-β2GPI, aPS/PT.

## Data Availability

The data presented in this study are available on request from the corresponding author. The data are not publicly available due to ethical reasons.

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
