# Peer review of "Antiphospholipid Antibodies and Vascular Thrombosis in Patients with Severe Forms of COVID-19"

_biomedicines, 2023, doi:10.3390/biomedicines11123117_

Round 1

Reviewer 1 Report

Zlatković-Švenda and colleagues present a study investigating for associations between aPL and thrombotic complications in patients with severe COVID-19. The design is largely sound but I have one major criticism, which is omission of lupus anticoagulant testing since it is the aPL that has the highest association with thrombotic and obstetric events in APS. The sentence beginning on line 42 acknowledges that aPS/PT have been shown to have a strong association with LA activity and clinical manifestations of APS, but this is insufficient evidence to justify exclusion of LA testing from a study investigating links between aPL and thrombosis in this patient cohort. aPS/PT are not yet criteria antibodies, and patients with LA but who are negative for aPS/PT have been described. Furthermore, LA tests are functional assays that detect LA activity irrespective of epitope specificity and the aPL profile is incomplete without them. Given that it is unlikely the patient samples can be re-visited for LA testing, the discussion needs to include a robust justification for exclusion of LA testing, clear acknowledgement that aPS/PT assays are not direct surrogates for functional LA assays, and why, and detailed consideration of implications for the outcomes and interpretations of the study data arising from this exclusion. The anticoagulation for COVID-19 patients can certainly interfere with LA testing, although there are strategies that can reduce or overcome their effects in some samples.

It is common to refer to aPL detected in solid-phase assays as aPL but consider LA separately, which is inaccurate because LA are also aPL. With this in mind, the title of the paper needs to be altered to be clear which aPL were tested.

Minor comments

Line 59 – in what way do the authors consider COVID-19 to be a coronary pandemic?

Line 70 – lactate dehydrogenase does not need to be italicised

Section 2.3 – only abbreviations of test names are required because they are defined earlier in the text.

Is ‘Eppendord Protien Low Bind Tubes’ meant to be Eppendorf Protein LoBind tubes? Manufacturer details should be given. Although the aPL ELISA assay protocols have been previously published, it would valuable to confirm that they were performed in accordance with international guidelines, which should be cited (J Thromb Haemost 2014;12:792-5), but only if it is actually the case. One cause for concern is that 99th percentile cutoffs for the three isotypes of each of aCL, aβ2GPI, and aPS/PT are identical. This is statistically unlikely and suggests there may have been some rounding/harmonisation. Please describe how these identical cut-offs were derived.

In Table 1, reference ranges must be given for all parameters in the legend, except the aPL as they have been given in the main text. LDH should be defined in the legend. ‘rate’ must be added after erythrocyte sedimentation.

In Table 1., total leucocytes, neutrophils and lymphocytes are reported in the same units (x 109/L) yet neutrophil and lymphocyte numbers far exceed total leucocyte counts. Could it be that the neutrophil and lymphocyte figures are actually percentages? If so, they should be converted to absolute numbers.

Typographical errors in Table 1: erytrocyte should be erythrocyte, haematocrite should be haematocrit, neutrophiles should be neutrophils

Section 3.2 states that there were significant decreases in leucocyte number and platelets at release and follow up compared to admission, but according to Table 1., leucocyte number and platelets were higher (from mean and mean +SD) at release and follow up. The slight increases in erythrocyte number, haemoglobin, and haematocrit at follow-up are described as improvements, yet they are little different from a clinical perspective, and are best described as such. Procalcitonin levels are described as unchanged but the means are 10 times lower at follow-up compared to admission. This comment is difficult to fully interpret without the local cutoff because cutoffs for indication of systemic infection vary, and some methods would consider the levels at admission abnormal but not the levels at follow-up.

aPL should be defined in the legend of Table 2.

The phrase ‘triple positivity’ in the context of APS specifically refers to positivity for aCL and aβ2GPI of the same isotype, plus LA, so it is inappropriate to use it in the discussion to describe positivity for aCL IgG, aβ2GPI IgM and aPS/PT IgM in one of the patients with COVID-19 pneumonia and arterial thrombosis.

The manuscript should expand on the potential significance of patients with a history of thrombosis having a significantly higher transient increase in aPL at discharge. Presumably this refers to increased levels rather than an increase in the numbers of assays with increased levels, which should be clarified.

Non-aPL related immunothrombosis in the pathogenesis of acute respiratory distress syndrome in COVID-19 is described in the literature so the promoting impact on haemostasis (not just coagulation) is not mainly unknown.

Although associations with elevated aPL and severe COVID-19 have been previously described, they are merely correlations and do not prove causality. Given that the aPL in most patients are transient, the elevated levels in severe COVID-19 more likely represent worsening of the disease state itself, and the concomitant further elevation of epiphenomenal antibodies is not necessarily remarkable. Since large numbers of patients were affected worldwide, it is inevitable that some already had APS, or had aPL that were destined to progress to APS, as was seen with one patient in this study. These considerations should be added to the discussion.

I agree that a major strength of the study is longitudinal clinical and laboratory data.

Just a few typographical errors

Author Response

REVIEWER ONE

Comments and Suggestions for Authors

Zlatković-Švenda and colleagues present a study investigating for associations between aPL and thrombotic complications in patients with severe COVID-19. The design is largely sound but I have one major criticism, which is omission of lupus anticoagulant testing since it is the aPL that has the highest association with thrombotic and obstetric events in APS. The sentence beginning on line 42 acknowledges that aPS/PT have been shown to have a strong association with LA activity and clinical manifestations of APS, but this is insufficient evidence to justify exclusion of LA testing from a study investigating links between aPL and thrombosis in this patient cohort. aPS/PT are not yet criteria antibodies, and patients with LA but who are negative for aPS/PT have been described. Furthermore, LA tests are functional assays that detect LA activity irrespective of epitope specificity and the aPL profile is incomplete without them. Given that it is unlikely the patient samples can be re-visited for LA testing, the discussion needs to include a robust justification for exclusion of LA testing, clear acknowledgement that aPS/PT assays are not direct surrogates for functional LA assays, and why, and detailed consideration of implications for the outcomes and interpretations of the study data arising from this exclusion. The anticoagulation for COVID-19 patients can certainly interfere with LA testing, although there are strategies that can reduce or overcome their effects in some samples.

It is common to refer to aPL detected in solid-phase assays as aPL but consider LA separately, which is inaccurate because LA are also aPL. With this in mind, the title of the paper needs to be altered to be clear which aPL were tested.

 ANSWER:  Thank you for pointing this out. We agree that assessing LA activity is important when examining the relationship between aPL and thrombosis, and that testing solid phase aPL (aCL, beta, and aPS/PT) cannot replace this testing completely. The exclusion of LA testing from our study is therefore a limitation of the study, however in the hospital setting the testing for LA activity with strategies that could reduce the effect of anticoagulant could not be performed in our case.

First, all of our patients were on anticoagulation therapy already due to COVID-19 pneumonia (low molecular weight heparin - Fraxiparine 2x0.6mL- 2x0,9mL daily, depending on the level of d-dimer). Although it is assumed that LMWH given with neutralisers would not affect the LA testing, the neutralizers are effective only up to specified LMWH levels (0.8–1.0 U/mL) that usually cover prophylactic doses only.

Second, all of the blood samples in our study were frozen and sent on dry ice to Slovenia, so that they could be done at the same place in order to be comparable.  Some studies have shown that the dry ice may affect sample pH and increase the fraction of false positive LA results. Third, all of our patients had an increase in C-reactive protein (CRP) due to COVID-19 pneumonia (mean (SD) 102.8 (72.3) mg/L). CRP interferes with PL in the reagents of the PL-dependent clotting assays used for LA. Elevated levels of CRP may prolong clotting times, resulting in false positive LA results.

According to the reviewer's advice, we have completely rewritten the introduction and discussion, added explanation for exclusion of the LA testing, and also added literature research that supports the use of aPS/PT, lines 45-52 and lines 232-253.

Minor comments

Line 59 – in what way do the authors consider COVID-19 to be a coronary pandemic?t

ANSWER: Thank you! This is typing error, it should be the worldwide pandemic. We have changed this in the manuscript.

Line 70 – lactate dehydrogenase does not need to be italicised

ANSWER: Thank you! We have corrected this in the manuscript.

Section 2.3 – only abbreviations of test names are required because they are defined earlier in the text.

ANSWER: Thank you! We have corrected this in the manuscript.

Is ‘Eppendord Protien Low Bind Tubes’ meant to be Eppendorf Protein LoBind tubes? Manufacturer details should be given.

ANSWER: Thank you! We have corrected this in the manuscript.

Although the aPL ELISA assay protocols have been previously published, it would valuable to confirm that they were performed in accordance with international guidelines, which should be cited (J Thromb Haemost 2014;12:792-5), but only if it is actually the case. One cause for concern is that 99th percentile cutoffs for the three isotypes of each of aCL, aβ2GPI, and aPS/PT are identical. This is statistically unlikely and suggests there may have been some rounding/harmonisation. Please describe how these identical cut-offs were derived.

ANSWER: Thank you for your comment. All three ELISAs (aCL, aB2GPI, and aPS/PT were established following recommendations and guidelines (J Thromb Haemost 2014;12:792-5), and this reference was added in the literature. Cut-off values were established at 99th percentile of >250 healthy control serum samples. Standards and calibrators (patients’ positive sera) for different isotypes G/M/A are differently diluted to reach the same cut-off in arbitrary units. All methods are regularly verified to determine between-run and within run variability and lot to lot variability.

The following text was added to the manuscript in line 120-126: ‘’ aPLA were determined at the Department of Rheumatology, University Medical Centre Ljubljana (UMCL), Slovenia, using in-house enzyme-linked immunosorbent assays (ELISA) according to predefined previously established protocols [23,24] in compliance with international guidelines protocol. Briefly, aCL were measured according to the protocol first described in 1997 (Bozic B, Int Arch Allergy Immunol. 1997;112(1):19-26) and later repeatedly evaluated (Avcin T, Cephalalgia : an international journal of headache. 2004;24(10):831-7.; Zigon P, J Immunol Res. 2015;2015:975704). anti-β2GPI were measured by in-house ELISA (Cucnik S, ClinChemLab Med. 2000;38(8):777-83) and evaluated by the European Forum for aPL (Reber, ThrombHaemost. 2002;88(1):66-73). aPS/PT were measured with in-house ELISA first described in 2011 (Zigon, ClinChemLab Med. 2011;49(6):1011-18) and later repeatedly evaluated (Zigon P, Clinical and Developmental Immunology.2013;2013:8.; Zigon P, Clinical rheumatology. 2019;38(2):371-8.)’’

In Table 1, reference ranges must be given for all parameters in the legend, except the aPL as they have been given in the main text. LDH should be defined in the legend. ‘rate’ must be added after erythrocyte sedimentation.

ANSWER: Thank you! We have added reference ranges for the blood test numbers and biochemistry in the table. We have added ‘’rate’ after erythrocyte sedimentation. We have defined LDH in the legend.

In Table 1. total leucocytes, neutrophils and lymphocytes are reported in the same units (x 109/L) yet neutrophil and lymphocyte numbers far exceed total leucocyte counts. Could it be that the neutrophil and lymphocyte figures are actually percentages? If so, they should be converted to absolute numbers. Typographical errors in Table 1: erytrocyte should be erythrocyte, haematocrite should be haematocrit, neutrophiles should be neutrophils

ANSWER: Thank you! You are correct the neutrophils and lymphocytes were expressed as percentages. We have converted them to absolute numbers and calculated statistics again. Thank you for noticing typographical errors. We have corrected them.

Section 3.2 states that there were significant decreases in leucocyte number and platelets at release and follow up compared to admission, but according to Table 1., leucocyte number and platelets were higher (from mean and mean +SD) at release and follow up. The slight increases in erythrocyte number, haemoglobin, and haematocrit at follow-up are described as improvements, yet they are little different from a clinical perspective, and are best described as such. Procalcitonin levels are described as unchanged but the means are 10 times lower at follow-up compared to admission. This comment is difficult to fully interpret without the local cutoff because cutoffs for indication of systemic infection vary, and some methods would consider the levels at admission abnormal but not the levels at follow-up.

ANSWER: Thank you for punting this out. It was a mistake; therefore, we have changed the statement accordingly. Line 152 ’’Compared with admission, there was a significant decrease in ESR, CRP as well as ferritin, troponin and LDH levels, while numbers of leucocytes, neutrophils and platelets increased both at hospital discharge and at 3-months follow-up.’’

The reviewer is correct the slight increases in erythrocyte number, hemoglobin, and hematocrit at follow-up are described as improvements, yet they are little different from a clinical perspective. We have additionally explained this with the sentence; Line 160: ’’At 3-month follow-up improvements were only seen in the erythrocyte number, hemoglobin, hematocrit, and D-dimer thought from clinical perspective this difference is minor.’’

The levels of procalcitonin are indeed 10 times lower, but the difference is not statistically significant. We have now corrected the sentence on order to clear this out. Line 157: ’’ During hospitalization and at follow-up, procalcitonin level decreased but the difference was not statistically significant.’’

aPL should be defined in the legend of Table 2.

ANSWER: Thank you! We defined aPL.

The phrase ‘triple positivity’ in the context of APS specifically refers to positivity for aCL and aβ2GPI of the same isotype, plus LA, so it is inappropriate to use it in the discussion to describe positivity for aCL IgG, aβ2GPI IgM and aPS/PT IgM in one of the patients with COVID-19 pneumonia and arterial thrombosis.

ANSWER: Thank you. That is correct. We change this in discussion, line 266: ’’one developed cerebral artery thrombosis and had was positive for aCL IgG, anti-β2GPI IgM and aPS/PT IgM on admission…’’

The manuscript should expand on the potential significance of patients with a history of thrombosis having a significantly higher transient increase in aPL at discharge. Presumably this refers to increased levels rather than an increase in the numbers of assays with increased levels, which should be clarified.

ANSWER: Thank you for pointing this out. The statement refers to an increase in the numbers of positive assays, as in patient with history of thrombosis levels of different aPL increased above cut-off. The explanation in the manuscript is in discussion, line 309-312: This refers to an increase in the numbers of positive assays, e.g. for patients with a history of thrombosis, levels of different aPL increased above the cut-off. However, none of these patients experienced thrombosis while hospitalized.“

Non-aPL related immunothrombosis in the pathogenesis of acute respiratory distress syndrome in COVID-19 is described in the literature so the promoting impact on haemostasis (not just coagulation) is not mainly unknown.

ANSWER: Thank you. ’’mainly unknown was changed to „has been described“

Although associations with elevated aPL and severe COVID-19 have been previously described, they are merely correlations and do not prove causality. Given that the aPL in most patients are transient, the elevated levels in severe COVID-19 more likely represent worsening of the disease state itself, and the concomitant further elevation of epiphenomenal antibodies is not necessarily remarkable. Since large numbers of patients were affected worldwide, it is inevitable that some already had APS, or had aPL that were destined to progress to APS, as was seen with one patient in this study. These considerations should be added to the discussion.

ANSWER: thank you very much for this thoroughly explanation. We have added it in the  discussion, line 345-351:  ’’While 50–70% of COVID-19 patients tested positive for at least one aPL, according to certain investigations they did not experience any thrombotic events [47]. Despite the prior descriptions of connections between elevated aPL and severe COVID-19 [48], these observations are only correlations and do not prove causality [40]. Since the majority of patients' aPL are transient, the elevated levels in severe COVID-19 are more likely due to the illness condition worsening, and the concomitant further elevation of epiphenomenal antibodies is not necessary.

I agree that a major strength of the study is longitudinal clinical and laboratory data.

Thank you very much for all useful comments, observations and suggestions.

Comments on the Quality of English Language

Just a few typographical errors

Submission Date

10 August 2023

Date of this review

16 Aug 2023 18:42:22

Bottom of Form

© 1996-2023 MDPI (Basel, Switzerland) unless otherwise stated

Reviewer 2 Report

This is a well-written and potentially interesting article on antiphospholipid antibodies in COVID-19 patients. However, several problems have to be addressed to reconsider the final recommendation including the presentation of results and thoroughness of analysis.

1. Please delete the words: background, aim, methods, results, and conclusions, from the abstract. Line 30, what is the exact p-value? Conclusions should include valuable information for clinicians and practitioners.

2. Please use APLA as a short form of antiphospholipid antibodies throughout the whole manuscript.

3. Lines 47-51, citation? The introduction is rather chaotic, some paragraphs should be developed, some should be merged, and there should be more links between paragraphs, thus, please carefully check this section and improve it so that it would be much easier to read and follow.

4. The text associated with statistics needs to be improved since it lacks detailed information (normality test, mean/IQR, etc.).

5. How was the diagnosis of COVID-19 made in enrolled patients?

6. Table 1 (but also the Table 2). Please change p-values: 0.000 into <0.001, round these p-values to round those p-values that did not reach statistical significance to two decimal places, and mark all statistical significance p-values with *. Mean (SD) should be presented as mean ± SD (also in the table). How could you explain results where the standard deviation is greater than the mean? In this case, it is probably better to present the results as median and IQR.

7. Table 2. It would be interesting to compare APLA positivity in general (at least one positive of all 9 analyzed) between different groups, but also aCL (IgG/IgM/IgA), and the rest two. Additional information with comparisons (p-values) is required (also a small paragraph in the discussion section).

8. Line 185. SARS-CoV2 => SARS-CoV-2

9. Lines 226-227, Citation?

10. Lines 275-276. How?

11. The discussion section is disordered, therefore, I suggest highlighting the most important messages in the first paragraph and then, developing it by citing relevant articles and changing the order of paragraphs to build larger ones. However, it would be beneficial to expand this section in papers investigating COVID-19 and APLA, especially in the follow-up analysis, thus, the following article might be helpful (doi: 10.1016/j.thromres.2023.01.016).

Round 2

Reviewer 1 Report

The authors have responded comprehensively to the reviewer’s comments and I have just minor comments to make.

Lines 155-157 persist in referring to the minor changes in red cell count, Hb, Hct, and D-dimer as improvements, which remains misleading as they are clinically insignificant. A better description would be “The slight increases in erythrocyte number, haemoglobin, and haematocrit at 3-month follow-up were clinically insignificant.”

I can see in Table 1 that neutrophils and lymphocytes have been altered to absolute numbers as requested, but why have they been altered for CRP, D-dimer, troponin, and procalcitonin? The changes to procalcitonin levels no longer display the 10-fold difference between admission and follow-up. The re-labelling of the second row header from Mean (SD) to Mean ± SD/Median (IQR) now promises more parameters than it delivers. Is the updated labelling erroneous or are some data missing? The IQR in this label, plus some data in the rows are suffixed with † but the legend does not indicate what it means.

Table 2 should indicate numbers of patients positive with more than one antibody type.

Line 246 – elevated CRP specifically affects aPTT testing but not dRVVT.

Lines 270-271 – anti-2GPI IgM should be anti-β2GPI IgM

Line 280 – ‘still remains’ is a tautology so ‘still’ can be removed

Line 347 – remove ‘necessary’ and replace with ‘necessarily remarkable’

See main report

Author Response

Thank you for your time, efforts and useful advices that will lead to better scientific performance of our manuscript 

Reviewer 2 Report

I would like to thank you Authors for revising the manuscript. I had some comments that should be addressed before publication:

1. Line 30. Please change p<0.05 into the exact value of p, p = 0.041.

2. There is no conclusion in the abstract that might be valuable from the clinical point of view.

3. The introduction is still difficult to follow, therefore I suggest writing longer paragraphs by combining sentences, rather than putting only 1-2 sentences in a separate paragraph as I wrote previously.

4. Statistical analyses should be described in detail focusing on aspects of which tests were used in which situations.

5. Table 1. In case of p-value>0.05, please round the p-value to two decimal places. The same should be done for the entire manuscript to maintain uniformity. p value => p-value (also Table 2); 1.000 => 1.00, 0.940 => 0.94, etc. *p ≤ 0.05 => *p < 0.05

6. Table 2, worsening. 3 patients had positive aCL IgG, thus 30.0%, not 21.4%, also please check the rest of the results Please round the results to two decimal places.

7. I believe that a short paragraph describing patients with comorbidities (hypertension, hyperlipidemia, diabetes mellitus, angina pectoris) should be presented in the text focusing on the hospital course and specific comparisons in terms of aPLA presence.

8. The results in section 3.3.4 should be also addressed in the discussion taking into account the current literature.

Minor editing of English language required.

Author Response

Thank you very much for your useful advices, time and efforts made to give more comprehensivness and importance to our manuscript

Round 3

Reviewer 2 Report

I accept all corrections that have been made. However, I have some comments. In general, it is hard to read a manuscript with so many changes, better to highlight them with one colour.

I suggest changing: aCLA into aCL antibodies, throughout the manuscript (main text, keywords), since this form is more often noticed in the literature.

Please combine paragraphs from lines 65-77.

Line 129: anti-IL-6 => anti-interleukin(IL)-6

Line 153: standard deviation => standard deviation [SD]

Line 177: persons => patients 

aPLA positivity (9)1 => please change 0.00 into <0.001 

Please combine paragraphs from lines 285-304 together.

Author Response

Thank you for your review and useful comments.

All corrections that have been made before are now accepted. New changes are highlighted in the text.
